# Sustainable and Eco-Friendly Packaging Films Based on Poly (Vinyl Alcohol) and Glass Flakes

**DOI:** 10.3390/membranes12070701

**Published:** 2022-07-11

**Authors:** Iftikhar Ahmed Channa, Jaweria Ashfaq, Sadaf Jamal Gilani, Ali Dad Chandio, Sumra Yousuf, Muhammad Atif Makhdoom, May Nasser bin Jumah

**Affiliations:** 1Thin Film Lab, Department of Metallurgical and Material Engineering, NED University of Engineering & Technology, Karachi 75270, Pakistan; jaweriaashfaq72@gmail.com; 2Department of Basic Health Sciences, Preparatory Year, Princess Nourah Bint Abdulrahman University, Riyadh 11671, Saudi Arabia; sjglani@pnu.edu.sa; 3Department of Building and Architectural Engineering, Faculty of Engineering & Technology, Bahauddin Zakariya University, Multan 60000, Pakistan; sumra.yousafrm@gmail.com; 4Institute of Metallurgy and Materials Engineering, University of the Punjab, Lahore 54590, Pakistan; atif.ceet@pu.edu.pk; 5Biology Department, College of Science, Princess Nourah Bint Abdulrahman University, Riyadh 11671, Saudi Arabia; mnbinjumah@pnu.edu.sa; 6Environment and Biomaterial Unit, Health Sciences Research Center, Princess Nourah Bint Abdulrahman University, Riyadh 11671, Saudi Arabia; 7Saudi Society for Applied Science, Princess Nourah Bint Abdulrahman University, Riyadh 11671, Saudi Arabia

**Keywords:** food packaging, PVA, glass flakes, barrier properties, degradation

## Abstract

The majority of food packaging materials are petroleum-based polymers, which are neither easily recyclable nor ecologically friendly. Packaging films should preferably be transparent, light in weight, and easy to process, as well as mechanically flexible, and they should meet the criteria for food encapsulation. In this study, poly (vinyl alcohol) (PVA)-based films were developed by incorporating glass flakes into the films. The selection of PVA was based on its well-known biodegradability, whereas the selection of glass flakes was based on their natural impermeability to oxygen and moisture. The films were processed using the blade coating method and were characterized in terms of transparency, oxygen transmission rate, mechanical strength, and flexibility. We observed that the incorporation of glass flakes into the PVA matrix did not significantly change the transparency of the PVA films, and they exhibited a total transmittance of around 87% (at 550 nm). When the glass flakes were added to the PVA, a significant reduction in moisture permeation was observed. This reduction was also supported and proven by Bhardwaj’s permeability model. In addition, even after the addition of glass flakes to the PVA, the films remained flexible and showed no degradation in terms of the water vapor transmission rate (WVTR), even after bending cycles of 23,000. The PVA film with glass flakes had decent tensile characteristics, i.e., around >50 MPa. Increasing the concentration of glass flakes also increased the hardness of the films. Finally, a piece of bread was packaged in a well-characterized composite film. We observed that the bread packaged in the PVA film with glass flakes did not show any degradation at all, even after 10 days, whereas the bread piece packaged in a commercial polyethylene bag degraded completely. Based on these results, the developed packaging films are the perfect solution to replace commercial non-biodegradable films.

## 1. Introduction

Packaging is a fundamental component of food items since it maintains the quality of the foods by blocking environmental attacks, such as the diffusion of gases, and especially oxygen and moisture. For product stability and shelf-life extension, food packaging films must exhibit appropriate mechanical strength and barrier properties, preferably biocompatibility and non-toxicity [1]. Various materials like metals, crystals, plastics, and composites have historically been used for packaging purposes [2]. These existing packaging films are sufficient in terms of quality for keeping food safe; however, after the consumption of the food, all these packaging films end up in rubbish channels [3]. These sorts of packaging films need years to decompose, and meanwhile, these disposed packaging films are a main source of contamination and result in increased pollution, posing a significant threat to the natural environment [4]. A significant proportion of discarded packaging is composed of polyethylene plastic, which has negative impacts on the environment [5]. Such concerns have produced a rise in the improvement of biodegradable packaging materials that have qualities proportionate to those of plastic packaging materials [6]. This study aimed to develop a biodegradable and non-toxic polymer film for food packaging applications. The developed packaging films exhibited improved mechanical strength and flexibility while maintaining the freshness and taste of food items over a longer timescale. This may enhance the realistic usefulness of the developed film through decreasing food disasters by enhancing the shelf life of food. The quality of packaging films decides the shelf life of the encapsulated food [7,8]. A variety of biodegradable polymers are used in the food packaging industry, which include polylactic acid (PLA), polyimide (PI) [9,10], and natural polymers such as gelatin [11,12] and fruit extracts [13,14,15]. Most of these polymers have good film formability but possess poor barrier characteristics against the diffusion of moisture and oxygen. Poly (vinyl alcohol) (PVA) is a food-grade synthetic polymer that does not have excellent film formability but does have excellent barrier properties against the diffusion of oxygen [16]. Hence, we chose this polymer for this study. PVA may be velvety or whitish, bland, odorless, non-toxic, biocompatible, thermostable, granular or powdered, semi-crystalline, or straight manufactured polymer [17]. It has astonishing optical properties, an expansive dielectric quality, and an amazing charge capacity [8]. Its mechanical, optical, and electrical qualities can promptly be custom-made by doping with fillers [18]. PVA is easily available in different grades and molecular weights commercially, which opens up a variety of applications for its use [16]. In addition to that, PVA is one of the polymers that have excellent oxygen barrier properties that can be tailored by adjusting the thickness of the films and filler contents [19]. PVA has good mechanical as well as thermal stability and hence is suitable for packaging applications [20]. PVA is broadly utilized for food packaging applications due to its biodegradability, non-toxicity, great film-forming capacity, absorbency, prepared accessibility, and low processing cost [21]. Though PVA is extremely sensitive to moisture and completely soluble in water, this characteristic can be altered by adjusting the molecular weight and incorporating intrinsic crystallinity or crystalline fillers into it. Furthermore, to overcome the moisture sensitivity and make PVA suitable for packaging applications, inorganic fillers with a high aspect ratio can be incorporated [8]. The incorporation of inorganic fillers will not only reduce the moisture sensitivity but also enhance the overall packaging properties by reducing the free volume within the polymer, thereby reducing the penetration of gases. The higher the aspect ratio of the fillers is, the higher that the blocking effect against permeating gases will be. According to Z. W. Abdullah and Y. Dong, adding 5 wt% of HNTs (Halloysite Nanotubes) to PVA films can increase the water resistance by 48%. Their study also concluded that the moisture-resisting values increased from 10% to 70% when the HNT concentration was increased. In addition to that, the films showed an acceptable range of transparency, strength, and biodegradability [22]. In a similar study, the influence of nano-cellulose (NC) and Ag NPs on the physical, mechanical, and thermal characteristics of PVA nanocomposite films was investigated by M. S. Sarwar et al. [23]. When the PVA was filled with 8 wt% NC, the tensile strength increased from 5.52 MPa to 12.32 MPa. The prepared nanocomposite films’ mechanical qualities and antibacterial capabilities suggested that these might be used in packaging applications [23]. Z. Peng et al. proposed that creating antibacterial PVA films is also possible by adding TiO_2_ nanoparticles. The oxygen transmission rate, swelling ratio, and water vapor uptake ratio of the films containing 0.5%–7% TiO_2_ NPs were reduced by 0.4%–0.8%, 22.82%–81.79%, and 3.59%–10.7%, respectively, when compared with the pure PVA film. Hence, PVA films have a promising future in antimicrobial food packaging according to researchers [24]. The incorporation of nanosized fillers enhances the overall performance of PVA, including its thermal stability, strength, and bendability, without affecting the transparency and biodegradability of PVA. Additionally, the production of high-aspect-ratio nanoparticles and their addition to films is not a cost-effective process. Thus, the addition of nanoparticles may increase the processing cost of PVA/composite films. Therefore, either alternative cost-effective procedures may be developed for the production of high-aspect-ratio nanosized particles, or other cost-effective fillers must be explored to produce economical and suitable packaging films. One prerequisite for other cost-effective fillers is that they must have an aspect ratio along with the matching refractive index to PVA. By applying an aspect ratio, transparency, and cost-effectiveness filters, the choice of fillers is limited to a few materials, including micro-sized cellulose crystals [25], natural clay [26], and glass flakes [27]. Cellulose crystals have smaller aspect ratios, i.e., around 50, because of their shape, whereas nanoclay has an excellent aspect ratio, but its intercalation and exfoliation mainly depend on the polymeric matrix. Glass flakes are cost-effective, transparent, and possess high aspect ratios, i.e., up to 2000 [28]. Additionally, since glass flakes are compatible with most of the polymers, we selected glass flakes as the fillers for this study.

Glass flakes are micro-sized disks of glass, having the thinnest possible thickness and a high surface area. The ratio of surface area to thickness is referred to as the aspect ratio. Commercially available glass flakes usually exhibit an aspect ratio between 200 to 2000. This aspect ratio along with their inertness make the glass flakes the perfect choice for packaging films. The addition of flakes in films reduces the films’ gas/fluid permeability and retains their mechanical stability [29]. Glass flakes dispersed throughout the films help prevent water vapor and chemical solutions from entering. Glass flakes provide a thermal stability layer within the protective coatings, reducing the risk of the coating breaking and peeling due to heat or stress [30]. Glass flakes significantly raise the hardness of epoxy and polyester resin coatings, increasing surface wear resistance. The large particle size allows for a higher aspect ratio, which is useful for blocking a gas’s penetration. The addition of such large-aspect-ratio flakes in higher concentrations favors the perfect orientation within the films. In studies carried out by Scharfe et al. [27] and Channa et al., [29], glass flakes were incorporated into the polyvinyl butyral (PVB) matrix, and composite films were prepared and characterized. The work done by Scharfe et al. focused on the mechanical characteristics and surface smoothness of the films, and the work done by Channa et al. elucidated the gas permeability and surface characteristics, as well as the transparency of the films. It was found that the incorporation of glass flakes (25 vol%) into the PVB matrix resulted in a reduction of oxygen and moisture permeability by a factor of around 450. In addition to this, the films also maintained a transparency of around 85% [31]. Since PVA is intrinsically an oxygen blocker but is extremely sensitive to water, glass flakes seem to be ideally suited for PVA. These large-aspect-ratio fillers can be incorporated in the PVA matrix, where they can act as the main blocking agents against the diffusion of moisture as well as oxygen, thereby increasing the overall film’s performance and stability. The prepared films can also maintain the overall transparency since the refractive index (*n* at 550 nm = 1.48) of PVA perfectly matches with the refractive index (*n* at 550 nm = 1.54) of glass flakes [29,32,33].

The purpose of this study was to create perfect, transparent, eco-friendly packaging films using a completely biodegradable and cost-effective PVA matrix through incorporating commonly available glass-flake fillers. Both materials are compatible with each other and have almost perfectly matching refractive indices. The composite of PVA/glass flakes has excellent processability, and the films can be processed easily with minimum processing steps. Apart from that, the primary goal was to develop cost-effective ultra-barrier coatings that fulfill the requirements of various packaging applications, including food and pharmaceutical products. In this study, thin films were prepared by mixing these two materials. The prepared films were then studied thoroughly. These thorough studies included aspects such as its appearance, transparency, internal structure via SEM analysis, moisture permeation, and protective response to a piece of bread packaged in it. It was observed that these films performed much better than commercial polyethylene films. Based on the data, we expect that prepared packaging films should not only be economical but also sustainable, eco-friendly films that fulfill packaging requirements and pose no environmental threat, unlike polyethylenes.

## 2. Materials and Method

### 2.1. Materials

PVA was obtained from Aldrich Chemistry (Darmstadt, Germany) (average Mw 85,000–124,000 87–89% hydrolyzed) and was used without any further processing. Eckart GmbH (Hartenstein, Germany) provided synthetic glass in the form of flakes.

### 2.2. Processing

A 15 wt% PVA solution was prepared by dissolving PVA granules in de-ionized water. To produce a homogenized PVA solution, a hot plate (temperature around 100 °C) was used, with a stirring rate of 500 rpm. The formation of a clear solution typically took 3–4 h. After a clear solution was obtained, glass-flake powder was added to the solution with different weight percentages, i.e., up to 30 wt%, and the mixture was mixed via a mechanical tumbler. To remove the trapped air, the mixed solution was placed in a vacuum desiccator for about 20 min while maintaining a vacuum of around 0.1 bar. When the solution was ready, the films were prepared on a PET substrate using a blade coater. All the films were prepared in ambient conditions and by setting a gap of 1000 µm on the applicator. The coating speed was also kept constant, i.e., 30 mm/s. The prepared films were placed in an oven (SS-00AB from MTI Corporation, Richmond, VA, USA) at a temperature of 60 °C for a few hours. When the films dried completely, they were peeled off of the PET and were used as free-standing films for all of the characterizations and tests.

### 2.3. Film’s Characterization

#### 2.3.1. Scanning Electron Microscopy

The surface morphology was monitored using a JEOL JSM-7610F Akishima, Japan scanning electron microscope (SEM), and secondary electron signals were used for acquiring high-resolution topographical images. The SEM was operated at 2 kV accelerating voltage, and the low probe current mode was set to 65 nA. The films were fractured in the liquid nitrogen environment for a cross-section analysis.

#### 2.3.2. UV-Vis Transmittance

The ultraviolet-visible spectra of the produced thin film were determined using a double beam UV-Vis spectrometer (Pharma Spec UV-1700, Shimadzu, and Kyoto, Japan). The absorbance’s wavelengths ranged from 200 nm to 800 nm.

#### 2.3.3. WVTR

Thwing-Albert Instrument Company (West Berlin, NJ, USA) supplied an ASTM E-96-compliant standard aluminum cup with a diameter of 6.35 cm. The experiment was carried out in accordance with the approach described by Channa et al. [34].

#### 2.3.4. Bending

A cyclic bend tester was employed. The tester had one fixed end, while the other end moved forward and backward to produce a radius of predefined value. Each film was subjected to various bending cycles, and the sample was analyzed for WVTR after every bending cycle set. This test determines the film’s stability against bending. The sample size for this test was 3 × 10 cm^2^, and after the test, a sample of 3 × 3 cm^2^ was cut from the middle of the bent sample for the permeation test.

#### 2.3.5. Tensile

The tensile testing of the films was performed in accordance with ASTM D882, and the testing was carried out using a Z005 Zwick/Roell universal testing machine (Ulm, Germany) at a fixed crosshead speed of 5 mm/min and a grip spacing of 50 mm. Rectangular samples with dimensions of 30 mm width and 150 mm length were required for tensile testing. A 5 N load was employed for the test.

#### 2.3.6. Hardness

The hardness of the nanocomposite was determined using a nanoindenter (Anton Paar nanoindentation hardness tester (Graz, Austria) with a diamond indenter). Poisson ratios of 0.45 were employed for PVA. The indentation was done under linear loading, with maximum loads of 10.00 mN while loading, and unloading rates were the same, i.e., 20.00 mN/min.

## 3. Results and Discussion

### 3.1. Morphological and Cross-Sectional Analysis

In this SEM study, glass flakes of 200 AR were used as filler particles to form a flexible barrier layer with increased permeability. Glass flakes were chosen as the filler particles because they have a number of advantages in terms of simple and efficient permeability. The optical micrograph of the flakes is shown in Figure 1a, where the size distribution clearly shows that the flakes are almost of the same size, and the size distribution did not seem to strongly impact or limit the results. The uniform dispersion of the glass flakes in the PVA film is shown in Figure 1b. They had a high aspect ratio and could easily be adjusted parallel to the film’s surface during the coating process due to their significant lateral expansion, which was in the same range as the film thickness, as shown in Figure 1c. The top view of the films suggested that the flakes were particularly positioned parallel to the film’s substrate, and Figure 1b confirms that. Based on the orientation of the flakes, the barrier film was expected to act as a suitable barrier against the diffusion of gases since the distribution of flakes within the matrix was more or less uniform [33].

### 3.2. Transparency

Figure 2 illustrates the UV-VIS spectra of a PVA filled with glass flakes. In the visible zone, pristine PVA had a transparency of 89%. With the addition of glass flakes, there was a modest increase in transmittance, which was attributable to a decrease in the reflectance of the pure PVA [32]. The addition of the glass flakes did not decrease the transparency since the refractive index of the PVA and the refractive index of the glass flakes matched perfectly with each other, i.e., 1.48 for PVA and 1.54 for the glass flakes [33,34,35,36]. Furthermore, when the filler quantity was increased, the transmittance of the layers decreased linearly [37]. The layers containing 5% and 15% fillers had transmittances of 90%. With 25% filler added to the PVA, the transmittance was reduced to 87%. This loss of transmittance can be attributable either to the light’s scattering caused by micronized glass flakes inside the film, or to any internal defect. The latter appears to be the most apparent possibility, given the transmittance fell linearly as the filler material increased [34,38]. These obtained results clearly demonstrate that the films were perfectly transparent and can be utilized as transparent food packaging for the encapsulation of dried fruits and bakery items.

### 3.3. WVTR

The pristine PVA was not an effective moisture barrier, with a WVTR of 22.5 g m^−2^ day^−1^, and the continuous increase in the concentration of the glass flakes created an extended route, making the moisture molecules take longer to diffuse; hence, the moisture permeability was reduced. The moisture permeation was 6.2 g m^−2^ day^−1^ to 1.2 g m^−2^ day^−1^ in the PVA layers with glass-flake concentrations of 5–25 vol%, respectively. When compared with a pristine PVA coating, this resulted in an 80% reduction in moisture permeability. The permeation decreased linearly as the concentration of glass flakes increased in the layer.

These experimental findings were then compared with Bhardwaj’s [39] theoretical permeation model. The experimental data were compared with theoretically estimated values. The tortuosity factor, layer thickness, and aspect ratio, as well as the order parameter based on the particles’ orientation inside the polymer matrix, were all computed using Bhardwaj’s model. According to Bhardwaj et al. [39], the order parameter (S) is an average orientation of the filler and is estimated using Equation (1) to be *S* = 0.80 [40].
(1)$$S=\frac{1}{2}\left(3cos^{2}\theta -1\right)$$

Equations (2) and (3) are used to compute the theoretical and experimental relative permeability of the films, and the results are shown in Figure 3.
(2)$$Relative\;Permeability_{\left(Bhardwaj\right)}\left[\;P_{c}/P_{P}\right]=\;\;\;\frac{1-{Ø}_{s}}{1+\frac{L}{2W}{Ø}_{s}\left(\frac{2}{3}\right)\left(S+\frac{1}{2}\right)}$$
(3)$$Relative\;Permeability_{\left(Experimental\right)}\left[\;P_{c}/P_{P}\right]=\frac{Permeability\;\left(composite\right)}{Permeability\;\left(polymer\right)}$$
where ø_s_ is the volume percentage of the fillers in the composite, and *L* and *w* are the length and width, respectively, of the fillers (aspect ratio). Figure 3 depicts a comparison of our experimental values to Bhardwaj’s theoretical model. These results were calculated using Equations (2) and (3). The experimental results were created and compared with different order parameters, namely, *S* = 0 (random orientation), *S* = 0.8 (calculated orientation from Figure 1c), and *S* = 1 (perfect orientation) [41]. The results reveal that the experimental results are in good agreement with the order parameter *S* = 1, indicating parallel orientation, and that they function as a good barrier in agreement with the average order parameter calculated from Figure 1c. As a result of this comparison, we confirmed and validated the barrier quality of the PVA filled with glass flakes as an order parameter of 0.8, with the parallel orientation calculated from the SEM cross-section [29].

### 3.4. Bending

Barrier materials must also be flexible without being harmed in order to encapsulate various foods. Thus, glass-flakes-based films should not be fragile and rigid. Hence, a bending test was performed to optimize the number of glass flakes and their impact on flexibility. The bending test was performed on films containing different amounts of glass flakes. The bending results are shown in Figure 4. The bending test was performed with a bending radius of 6 cm. Figure 4 clearly shows that increasing the amount of the glass flakes decreased the flexibility, as the initial WVTR of the bent films deteriorated. Figure 4 depicts a comparison of the bending performance of the films [34]. The pristine PVA films retained their initial WVTR values after 20 K bending cycles. Even after 10 K bending cycles, the water-barrier characteristics of the PVA films with glass flakes were reduced by 5 to 25% of their initial value. This was because the PVA at a film thickness of 100 µm remained flexible and showed no adverse effects of bending in terms of WVTR. The films having 5 wt% glass flakes exhibited a slight decrease (around 2%) in the films. Similarly, the films having 15 wt% and 25 wt% of glass flakes also showed a decreasing trend in bendability. A decrease of around 3% and 5% was observed in the films having 15 wt% and 25 wt% of glass flakes, respectively [42]. The decrease in WVTR was not that significant, which meant that the glass flakes and PVA had a firm adhesion to each other. Due to the firm adhesion between the glass flakes and the PVA, when the films were bent, the glass flakes were disoriented but returned to their original orientation when the forces were removed. A small decrease in WVTR may be due to the generation of nanosized defects, while the bending allowed moisture to decrease and caused overall permeability in the film [42].

### 3.5. Tensile

The mechanical properties of the PVA and PVA film with glass flakes were determined using tensile strength (TS) and tensile elongation (TE). The results of tensile test are shown in Figure 5, and the corresponding values are mentioned in Table 1. Table 1 shows the TS and TE of all the samples that were examined. The PVA had a TS and TE value of 55 MPa and 5.5 %, respectively. Because of the strong crystalline structure of glass flakes, these values were smaller than the PVA with glass flakes. The PVA’s structure caused a well-formed intermolecular network to develop between its chains, resulting in good mechanical characteristics [43].

The addition of glass flakes to the PVA, on the other hand, decreased the TE and increased the TS value due to slightly inferior interfacial bonding between the PVA and glass particles. A possible cause for this behavior could be the presence of -OH bonds between the PVA and glass flake surface. The presence of free -OH contributed to the slightly inferior bonding between the PVA and glass flakes. However, this inferiority did not severely affect the overall mechanical strength of the films, since a drastic decrease in the tensile properties was not witnessed. The decrease in the TE values of the films containing flakes could be associated with the internal defects. A possible cause of the formation of internal defects could be the fast evaporation rate of the solvent during the drying period [44]. This could also be the reason that the WVTR of the PVA films with glass flakes slightly decreased with an increase in flake concentration [29].

### 3.6. Hardness

The mechanical characteristics of the PVA produced were investigated using the nanoindentation method, assuming that the PVA was nanometric. The Young’s modulus and hardness of the PVA–GF films were calculated using the PVA and glass flakes films’ penetration depth vs. time and load vs. displacement curves (Figure 6a,b). PVA film exhibits the typical curve of a soft polymeric material through which it can show more penetration with time, compared with other films in which glass flakes are added. The addition of the flakes decreased the softness of the PVA, exhibiting dissimilar behavior but maintaining the trend of pure polymer as shown in Figure 6a. Another point to be noted is that, compared with the other systems, the penetration depth with the glass flakes integration was reduced with regard to force, as shown in Figure 6b. As force was applied to the PVA film, which was soft and flexible, there was deep penetration when compared with the film including the glass flakes because the glass flakes were brittle, which reduced their softness and was the main reason that the indentor could not penetrate any further. This also proved that the addition of flakes in soft PVA could drastically enhance the overall hardness of the film [45]. The high instrumental elastic modulus (EIT = 1.49 GPa) and instrumental hardness measurements demonstrated a considerable increase in the hardness of the PVA following the addition of the glass flakes [46,47].

### 3.7. Degradation

The biodegradation study was carried out on bread pieces packaged in low-density polyethylene (LDPE) bags and prepared PVA films with 15% glass flakes. Bread pieces were encapsulated in glass-flakes-based PVA films and commercial LDPE films. The encapsulation process was carried out in an inert atmosphere using a glove box. The encapsulated samples were placed in ambient conditions in the room for 10 days. The degradation data are shown in Figure 7 and Figure 8. Figure 7 shows the degraded polyethylene bag packaged bread piece sample. We observed that the bread piece started to degrade by means of fungal growth at some point after day 3. With the passage of time, the fungus grew all over the piece, and the quality of the piece was deteriorated due to low oxygen and the high carbon dioxide partial-pressure effects on a wide range of fungal species. This fungus growth mechanism is supported by the work done by Legan J.D. (2021) [48]. On the other hand, the degradation of the bread piece encapsulated in the PVA with glass flakes is shown in Figure 8. It was evident from the data that the bread piece maintained its quality and freshness even after day 10. No fungus growth was observed in the sample during the course of the testing period. Furthermore, the freshness of the bread piece also remained intact. These obtained results clearly indicate the PVA and glass-flakes-based coatings are ideally suited for the packaging of food items. The films remained flexible and kept the freshness of the food for a sufficiently long period, increasing the shelf life of the packaged food [49].

A five-day soil burial degradation test was also performed to examine the biodegradability of the developed films. In this test, the films were buried in the soil for about five days, and photographs were recorded. The results are presented in Figure 9. It is evident from the photos that about 40–60% of the weight of the original films was lost, which is an obvious indicator of the biodegradation process carried out by the combined effect of micro-organisms and the moisture present in the soil. The film made of PVA and glass flakes blended well with the soil and began to deteriorate. By day 5, around 40% of the PVA film with glass flakes had degraded, whereas the PVA film had almost broken into pieces, with around 60% of that sample already degraded. Comparing the PVA/glass-flakes-based bioplastics to synthetic plastic, this study found that PVA-based films deteriorated faster in moisturized soil.

## 4. Conclusions

Glass-flakes-based PVA films were developed, and the developed films exhibited perfect transparency (total transmission of around ~90% at 550 nm) in the visible region of the spectrum. In addition to that, the developed films showed decent barrier properties against moisture (80% improved barrier properties, compared with pristine PVA) in different conditions, and the results were in complete accordance with the theoretical permeability model proposed by Bhardwaj. Bhardwaj proposed that perfectly oriented flakes would produce an order parameter of (S) = 1, whereas the orientation of the glass flakes observed via SEM as a cross-section order parameter calculated from Bhardwaj revealed the value of S = 0.8. This was a perfect match between the experimental data of this study and the theoretical data. Furthermore, it was also observed that increasing the concentration of glass flakes enhanced the barrier characteristics against moisture, but it affected the flexibility as well as the transparency of the films. The films with the highest glass flake loadings showed a loss of around 5% both in barrier properties and transparency. This was further confirmed by the tensile values, which also showed a decreasing trend with increasing concentrations of the glass flakes. These optimizing data clearly indicated that the films with the 15% of glass flakes were the best in all aspects. Hence, these optimized films were employed for the degradation of the bread piece. From the degradation data, it was also shown that the PVA films with glass flakes were much better than commercially used LDPE for the packaging of bread items. The piece packaged in the PVA films with glass flakes not only resisted the diffusion of air but also kept the piece fresh and fungus-free for a period of around 10 days. Additionally, a soil test also showed the biodegradability of the films as the samples lost 60% of their initial weight only in 5 days. Based on the set of data presented in this work, we concluded that the PVA/glass-flakes-based films are much better than commercial polymers in terms of maintaining the food’s quality as well as keeping the environment safe when these films are disposed. LDPE takes years to decompose, while the PVA-based films are easily biodegradable and pose no threat to the environment.

## Figures and Tables

**Figure 1 membranes-12-00701-f001:**
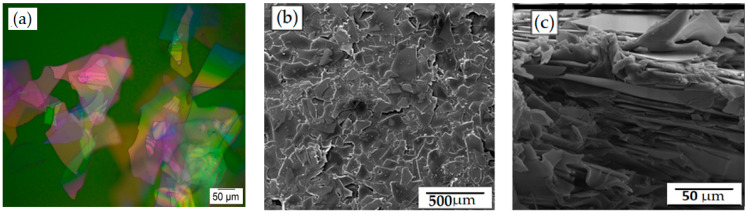
Micrographs of glass flakes and glass-flakes-based composites. (**a**) Optical micrograph of glass flakes shows (**b**) a top view SEM micrograph of PVA with glass flakes (25 wt%), demonstrating the size and dispersion of the flakes in the PVA matrix, whereas (**c**) shows a cross-section of the same film demonstrating a parallel orientation of the flakes within the PVA matrix.

**Figure 2 membranes-12-00701-f002:**
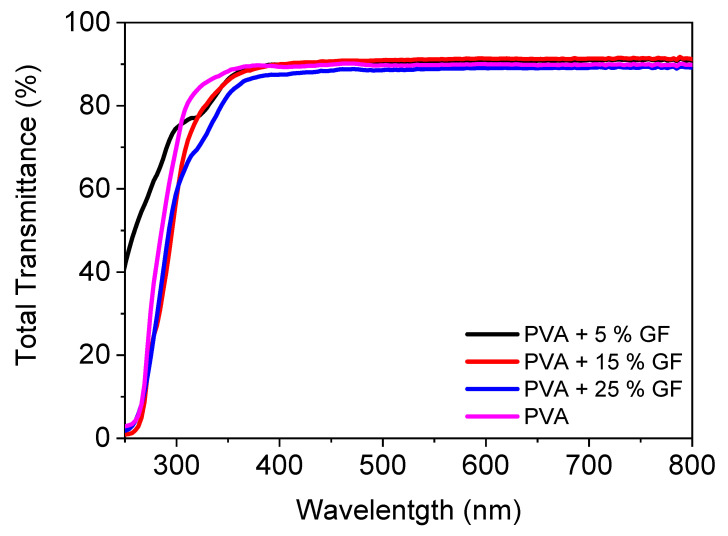
Total transmittance spectra of PVA films filled with and without 5–25 vol% of glass flakes of 200 AR.

**Figure 3 membranes-12-00701-f003:**
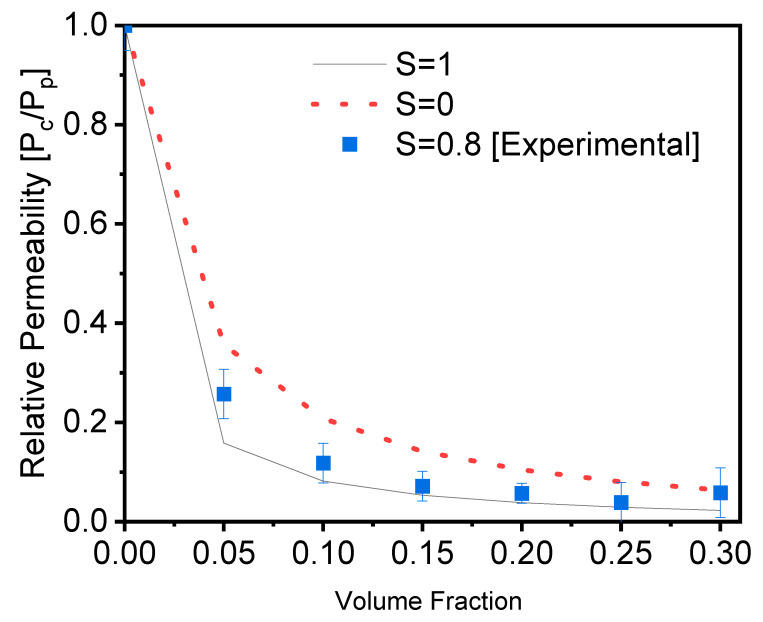
Evolution of the relative permeability of PVA films with an increasing content of glass flakes (square = experimental data, and dotted line = single line), according to the Bharadwaj model, for an aspect ratio of α = 200 and order parameters of S = 0, 1, and 0.8.

**Figure 4 membranes-12-00701-f004:**
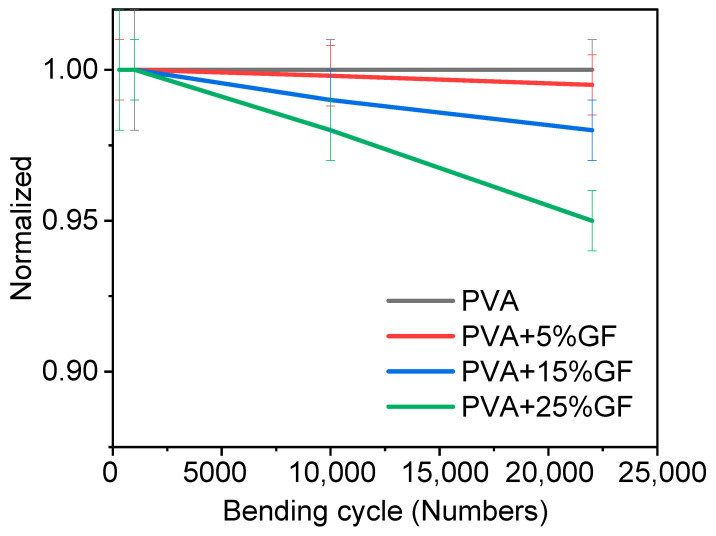
Normalized WVTR value of PVA film with glass flakes of 0 wt%, 5 wt%, 15 wt%, and 25 wt% plotted against the number of bending cycles at a bending radius of 6 cm. The black curve represents pristine PVA film, the red curve represents the PVA film with 5 wt% of glass flakes in it, the blue curve represents the PVA film with 15 wt% of glass flakes in it, and the green curve represents the PVA film with 25 wt% of glass flakes in it vs. the number of bending cycles. All the rested films had thickness of 100 µm.

**Figure 5 membranes-12-00701-f005:**
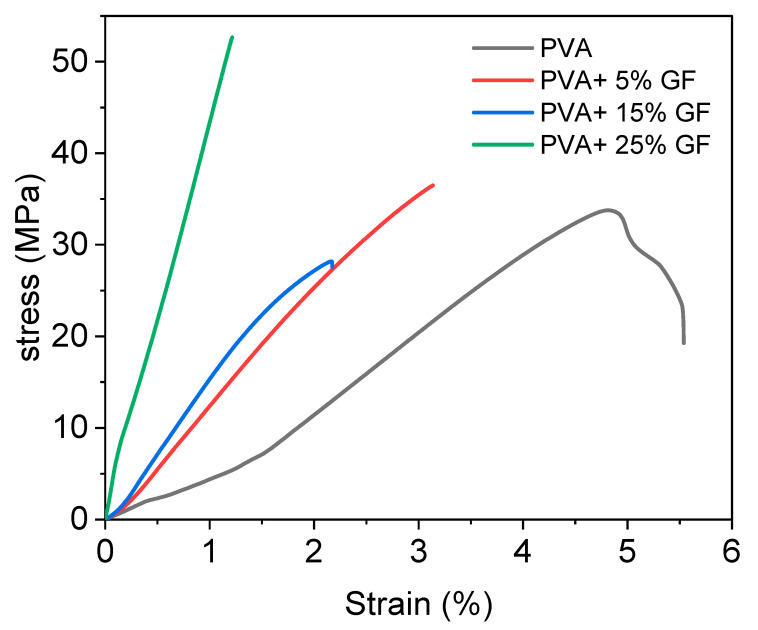
Stress–strain diagram of PVA film with different concentration of glass flakes (GF). The black line indicates the simple PVA film, the red line indicates the 5% GF with PVA, the blue line indicates the PVA with 15% GF, and the green line indicates 25% GF.

**Figure 6 membranes-12-00701-f006:**
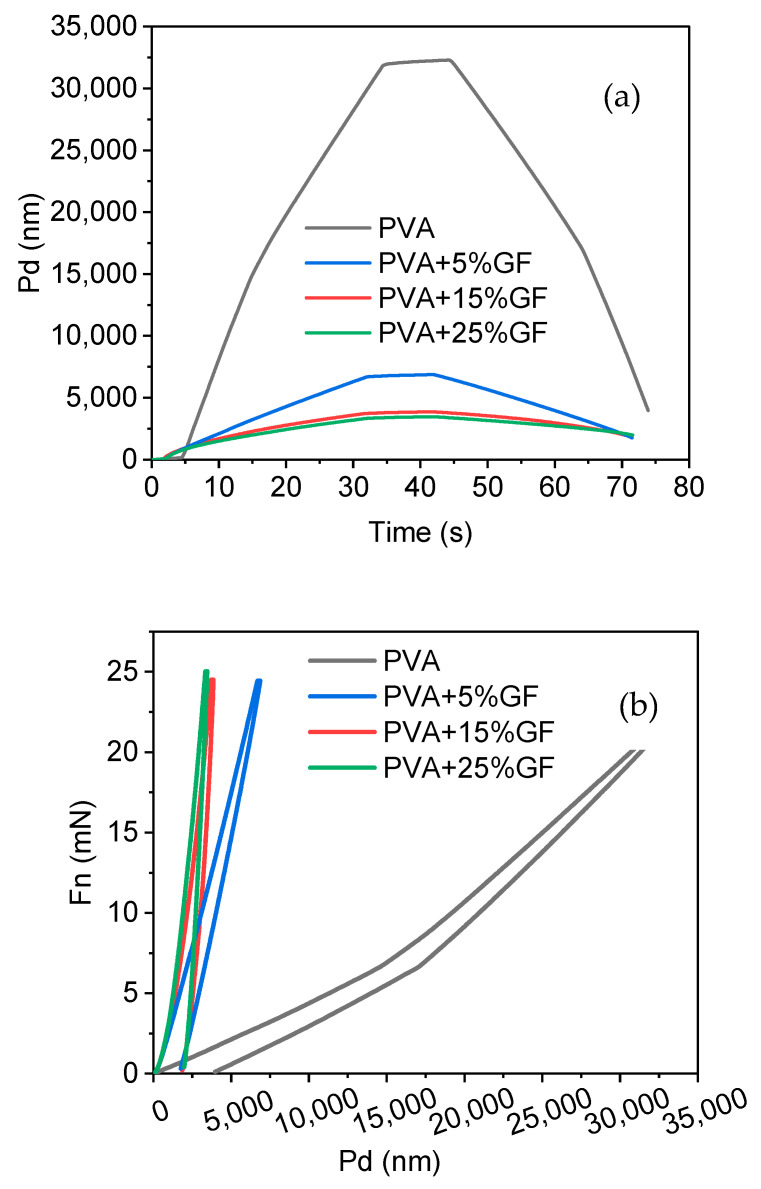
Nanohardness results. (**a**) Penetration depth vs. time curves of the PVA and PVA thin films with glass flakes (GF). (**b**) Force vs. penetration depth: PVA film indicated by the black line, PVA with 5% GF film indicated by the blue line, PVA with 15% GF film indicated by the red line, and PVA with 25% GF film indicated by the green line.

**Figure 7 membranes-12-00701-f007:**
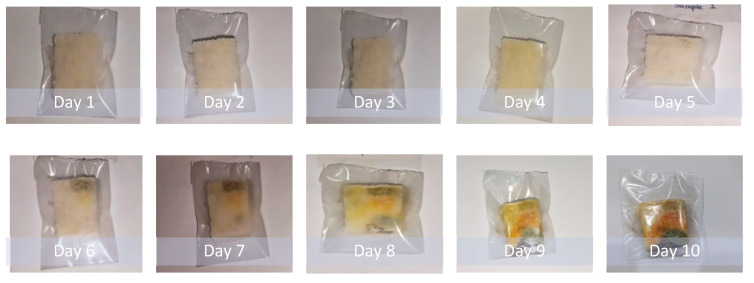
The bread piece was packaged in commercial polyethylene films and placed in an open atmosphere in ambient conditions (30 °C and 50% RH) for 10 days to see the effect of the packaging on the degradation of the bread piece.

**Figure 8 membranes-12-00701-f008:**
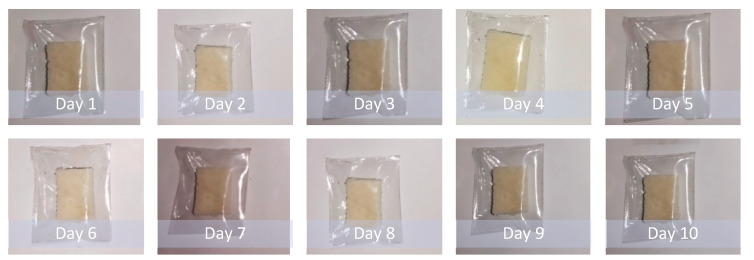
The bread piece was packaged in PVA with 15 % GF films. This sample was placed in an open atmosphere (30 °C and 50% RH) for 10 days to see the effect of the packaging.

**Figure 9 membranes-12-00701-f009:**
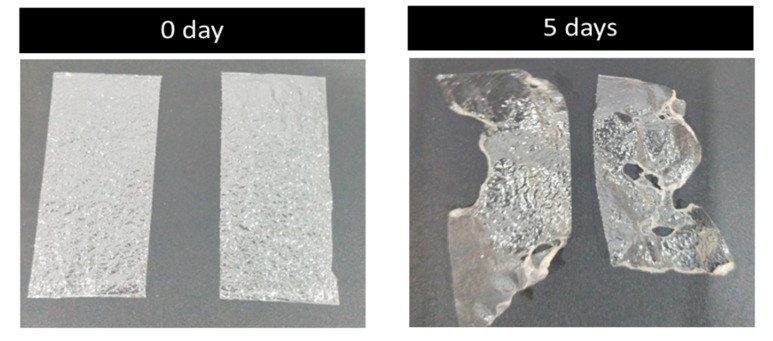
Biodegradability: soil burial degradation test of the films. The soil burial test samples before (0 day) and after the test (5 days).

**Table 1 membranes-12-00701-t001:** Tensile strength and tensile elongation of the PVA and PVA films with glass flakes.

S.No	Sample Code	Tensile Strength (MPa)	Tensile Elongation (%)
1	PVA	28	5.5
2	PVA + 5%GF	35	3.1
3	PVA + 15%GF	27	2.2
4	PVA + 25%GF	55	1.2

## Data Availability

The data presented in this study are available on request from the corresponding author.

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
