# Peer review of "Sustainable and Eco-Friendly Packaging Films Based on Poly (Vinyl Alcohol) and Glass Flakes"

_membranes, 2022, doi:10.3390/membranes12070701_

Round 1

Reviewer 1 Report

This revised version meets to the criteria to be published as article at Membranes.

Author Response

Many thanks for the reviewer for accepting our revisions in the manuscript

Reviewer 2 Report

Sustainable eco-friendly packaging films based on poly (vinyl alcohol) and glass flakes

Comments:

Abstract

Abstract is still long: Line 1-9: ‘Food….mechanically flexible.’ Can be moved to introduction

Line 23-25: The bread did not completely degrade, the images only shows the growth rate of mold

Introduction still not cover other studies related to the topic and why this study was carried out? I do understand the aim is to come up with eco-friendly packaging film but how it differs from other studies carried out under the same objective.

Materials and methods

2.2 Processing

Do we need the subtitle to be ‘Processing” or “Film formation of PVA and PVA/glass flakes”?

Overall: The manuscript need extensive English editing since it is difficult to follow what the authors are trying to address. Some of the sentences are long which also makes it difficult to follow. 

Author Response

Reviewer # 02

Comments:

Abstract

Abstract is still long: Line 1-9: ‘Food….mechanically flexible.’ Can be moved to introduction

Answer: Many thanks for highlighting this issue. Authors feel very sorry for this. In the revised manuscript, these lines are rephrased and shortened. We hope this is as per reviewer’s suggestion.

Line 23-25: The bread did not completely degrade, the images only shows the growth rate of mold

Answer: The authors are very thankful to reviewer for this valuable comment. In the degradation test, the theme was to show the protective action of the developed films. Our developed films protected the bread and did not let microorganisms to grow. While on the other hand, bread packed in commercial LDPE films showed the growth of mold. To prove the biodegradability point, an addition soil test analysis is added in the revised manuscript. The test clearly shows the action of moisturized soil, as the 60% of the films degraded within five days.

Introduction still not cover other studies related to the topic and why this study was carried out? I do understand the aim is to come up with eco-friendly packaging film but how it differs from other studies carried out under the same objective.

Answer: Many thanks to reviewer for highlighting the very important point. The introduction section is now updated and modified as per the valuable suggestion of the reviewer. All of the newly added information is highlighted in yellow in the revised manuscript.

Materials and methods

2.2 Processing

Do we need the subtitle to be ‘Processing” or “Film formation of PVA and PVA/glass flakes”?

Answer: The subtitle has been renamed as ‘Processing’ and Films formation …. PVA/glass flakes is removed in the revised manuscript.

Overall: The manuscript need extensive English editing since it is difficult to follow what the authors are trying to address. Some of the sentences are long which also makes it difficult to follow. 

Answer: Authors are extremely thankful to reviewer for this comment. The revised manuscript has now been thoroughly checked and corrected for such type of errors.

Reviewer 3 Report

Comment review

Packaging refers to materials used to protect products from packaging, facilitate storage and transportation, and promote sales. The research paper presents a biodegradable polymeric packaging film based on poly (alcohol) and glass flakes. Glass is made of inorganic minerals found in nature, so it is degradable. Micro-glass disks with high surface area and aspect ratio were mixed into biodegradable PVA materials to prepare biodegradable composite films. Glass flakes were added to enhance the vapor barrier property, gas barrier property, thermal stability and mechanical stability of the film. The addition of glass sheet makes the composite film have a certain orientation, and the transmittance of visible light of the composite material is not significantly affected due to the high transmittance of glass sheet. However there are still significant improvement need to make corrections. My detailed comments are as follows:

1.     More details on the materials should be provided, such as the grade, molecular weight, etc.

2.     The scale bar for the SEM images should be redrawn. In addition, the original morphology of glass flakes should be provided.

3.     One highly relevant review articles on the packaging materials should be cited: Nanomaterials 10 (1), 150, 2020.

4.     The expression of data should be in a scientific way with average value and standard error.

5.     In the degradation part, only the change of freshness of different series of packaging films in bread slices over time was studied, and the change of degradation of packaging films over time was not shown. Therefore, the change of degradation of packaging films over time should be shown in this part.

6.     Because of the uneven dispersion of glass sheet, it is easy to cause the problem of unstable mechanical properties, so the mechanical tensile curve is needed to display intuitively.

7.     Whether the addition of glass flakes will cause uneven surface of packaging materials or uneven thickness of materials due to uneven dispersion in the process of dispersion? It is necessary to record the thickness at different positions of samples and calculate the error bar.

8.     Because food packaging for internal packaging, so in the packaging process need to consider its health performance, in the process of transportation products will be subjected to extrusion, stretching, impact and other external forces, in the process of action whether micro glass will migrate to food.

9.     More property comparison with previous reports on packaging should be performed. Please carefully read the following highly relevant articles, compare and discuss in the manuscript: Development and characterization of food packaging bioplastic film from cocoa pod husk cellulose incorporated with sugarcane bagasse fibre; Packaging and degradability properties of polyvinyl alcohol/gelatin nanocomposite films filled water hyacinth cellulose nanocrystals; etc.

10.  There are some other syntax and formatting errors that need to be carefully checked again.

Author Response

Reviewer # 03

Packaging refers to materials used to protect products from packaging, facilitate storage and transportation, and promote sales. The research paper presents a biodegradable polymeric packaging film based on poly (alcohol) and glass flakes. Glass is made of inorganic minerals found in nature, so it is degradable. Micro-glass disks with high surface area and aspect ratio were mixed into biodegradable PVA materials to prepare biodegradable composite films. Glass flakes were added to enhance the vapor barrier property, gas barrier property, thermal stability and mechanical stability of the film. The addition of glass sheet makes the composite film have a certain orientation, and the transmittance of visible light of the composite material is not significantly affected due to the high transmittance of glass sheet. However there are still significant improvement need to make corrections. My detailed comments are as follows:

  1. More details on the materials should be provided, such as the grade, molecular weight, etc.

Answer: Authors are very thankful to reviewer for this comment.. and per suggestion, molecular weight is mentioned against the name of polymers in the revised manuscript.

  1. The scale bar for the SEM images should be redrawn. In addition, the original morphology of glass flakes should be provided.

Answer: Many thanks to respected reviewer, the SEM bar is redrawn and original morphology of the glass flakes is also shown in figure 1(a), and the relevant analysis is highlighted in the text.

  1. One highly relevant review articles on the packaging materials should be cited: Nanomaterials 10 (1), 150, 2020.

Answer: Extremely grateful to reviewers, this very interesting article has been cited in the revised manuscript.

  1. The expression of data should be in a scientific way with average value and standard error.

Answer: Please accept our apologies for the inconvenienced for the non-scientific expressions, however, in the revised manuscript, we have tried to be more specific and scientific while discussion the results. We hope, the revised version is in good accordance with the suggestions of the reviewers.

  1. In the degradation part, only the change of freshness of different series of packaging films in bread slices over time was studied, and the change of degradation of packaging films over time was not shown. Therefore, the change of degradation of packaging films over time should be shown in this part.

Answer: The authors are very thankful to reviewer for this valuable comment. In the degradation test, the theme was to show the protective action of the developed films. Our developed films protected the bread and did not let microorganisms to grow. While on the other hand, bread packed in commercial LDPE films showed the growth of mold. To prove the biodegradability point, an addition soil test analysis is added in the revised manuscript. The test clearly shows the action of moisturized soil, as the 60% of the films degraded within five days.

  1. Because of the uneven dispersion of glass sheet, it is easy to cause the problem of unstable mechanical properties, so the mechanical tensile curve is needed to display intuitively.

Answer: We are thankful to reviewers for their valuable comments. As per revised manuscript, the Figure 1b and Figure 1c show the SEM micrographs of the developed films, it is clearly evident from these results that the glass flakes are perfectly well distributed in the PVA matrix. However, with glass flakes loadings, some microlevel imperfections may generate, and cause the reduction in the elongation. However, tensile curves are added in the revised manuscript.

  1. Whether the addition of glass flakes will cause uneven surface of packaging materials or uneven thickness of materials due to uneven dispersion in the process of dispersion? It is necessary to record the thickness at different positions of samples and calculate the error bar.

Answer: We are thankful to reviewer for this comment highlighting very important issue in glass flake based coatings. Yes, we agree that the addition of microsized flakes may clearly cause roughness problem in the film. However, we have taken measures to deal with this issue. We tried our best to keep the surface roughness low while coating process. Furthermore, to eliminate thickness differences, we tested all of the films having same thickness i.e. 100 microns.

  1. Because food packaging for internal packaging, so in the packaging process need to consider its health performance, in the process of transportation products will be subjected to extrusion, stretching, impact and other external forces, in the process of action whether micro glass will migrate to food.

Answer: We are thankful to reviewer for this very valid comment. There are number of reasons for selecting glass flakes for this study. One of the most common reason is their non-reactivity and inertness towards human body along with its abundant availability. Furthermore, the adhesion of flakes with polymer matrix is very important. If the adhesion is excellent, therefore will be minimum chances of the flake to migrate into food. The adhesion of flakes with PVA matrix has been really great as can be seen in the result of bending. The films remain extremely bendable showing extra ordinary compatibility of PVA and flakes. Based on results obtained, we conclude that there are minimum chances of flakes migrating into food.

  1. More property comparison with previous reports on packaging should be performed. Please carefully read the following highly relevant articles, compare and discuss in the manuscript: Development and characterization of food packaging bioplastic film from cocoa pod husk cellulose incorporated with sugarcane bagasse fibre; Packaging and degradability properties of polyvinyl alcohol/gelatin nanocomposite films filled water hyacinth cellulose nanocrystals; etc.

Answer: Extremely thankful to reviewer for their kind suggestions and comments, more recent and highly relevant publications are now cited in the revised manuscript.

  1. There are some other syntax and formatting errors that need to be carefully checked again.

Answer: Many thanks to reviewers for their valuable comments and suggestions, the revised manuscript has been carefully read and corrected for formatting and syntax errors. We hope, the revised manuscript in now in good accordance with the comments and suggestions of the reviewers.

Round 2

Reviewer 2 Report

I recommend acceptance

Reviewer 3 Report

Authors have addressed all the previous issues well. An acceptance is suggested.

This manuscript is a resubmission of an earlier submission. The following is a list of the peer review reports and author responses from that submission.

Round 1

Reviewer 1 Report

This article deals with Biodegradable and environmentally friendly packaging films, but I didn't see any bio-fungicides or bio-bactericide effects or methods or results about the studied films, which is necessary to conclude the overall product.

Reviewer 2 Report

 Biodegradable and environmental friendly packaging films based on poly (vinyl alcohol) and glass flakes for the encapsulation of food

The study deals with the preparation of PVA reinforced with glass flakes for food packaging. This study is of interest when looking into the single-use plastics that need to be replaced by accessible and cheaper materials which are eco-friendly.

Comments:

Title and abstract are too long. Hence the authors have to shorten them to be catch so that it can attract readers.

Introduction

Pg 2 line 3: replace ‘expansion’ with ‘extension’

Pg 2 line 25: ‘semi crystal-line’ has to be ‘semi-crystalline’

Pg 2 line 28-29: ‘PVA…hydrolysis[8]’ has to rephrased so that it deliver the intended message.

Pg 2 line 30-31: ‘Adaptable,….properties’ has to be rephrased so that it deliver the intended message

Pg 2 line 31-32: ‘To avoid….from [10].’ It seems as if the sentence is not complete

Pg 2 line 32-33:’PVA….and food packaging.’ Need to be rephrased

Pg 2 line 36: replace ‘inexpensive cost’ with ‘inexpensiveness’

Pg 2 line 38-40: ‘To overcome…incorporated [7].’ Out of curiosity, how does aspect ratio of the filler has to do with moisture absorption?

Pg 2-3: line 40: ‘According to…..percent.’ HNTs were added into composite, i.e. PVA/ST/GL/HNT nanocomposite. The question will be why adding HNTs into the composite that has already have HNTs?

Pg 3 line 3: replace ‘increase’ with ‘increased’

Pg 3 line 9: Reference is missing

Pg 3 line 11-12: Remove ‘Furthermore, …over 90%’ since this sentence has nothing to do with packaging.

Pg 3 line 13-16: ‘Z. Peng et al……using TiO2. Has to be rephrased

Pg 3 line 21-22: The ratio….200 to 2000’ has to be rephrased

Pg 3 line 23-25: which….delivery [16]’ has to be rephrased

Pg 3 line 28-30: ‘Glass….wear resistance’ has to be rephrased

Pg 3 line 33-34:’This powder….porousness.’ has to be rephrased

Can authors incorporate some literature where glass flakes were used as filler for PVA or other polymeric materials for packaging applications? This will give the readers the sense of why this study was carried out rather than mentioning the properties of glass flakes.

Pg 3 line 37: ‘what is meant by ‘can never be manufactured’?

Pg 3 line 40-41: ‘The moisture….eliminate plastic’ need to rephrased

The objective or novelty of this study is not clear, hence the modification of the last two paragraphs can give readers idea of why the investigation was carried out.

2. Materials and methods

There are two ‘2.1’ i.e. for materials and for method

Under Method: What “Polymer barrier film without filler” has to do with the method?

Pg 4 section 2.3.1:

‘manufactured’ has to be removed

“at” is missing between operated and 2kV

Replace ‘broken’ with ‘fractured’

Pg 4 section 2.3.2:

Wave lengths has to be one word, i.e. wavelength

Spacing is missing between the numbers and the units

Pg 4 section 2.3.4:

Spacing is missing between ‘cm2’ and ‘and’

2.3.4. Tensile (check numbering again since it the same as for bending)

Line 1: replace ‘is’ with ‘was’

Line 4: Spacing is missing between the numbers and the units

Results and discussion

Section 3.1:

Figure 2(a) and Figure 2(b) supposed to be Figure 1(a) and Figure (b).

Line 4: The GF filler….in Figure 2(a). has to be rephrased

Line 9: ‘f’ for figure has to be in uppercase

Line 9: spacing is missing between ‘suitable barrier’ and ‘[19]’

Section 3.2:

 Spacing is missing between the numbers and the units

Pg 6: sentence “These….applications’ has to rephrase

Figure 2 spacing is missing ‘filled with’ and ‘and’

Section 3.3:

Pg 7: ‘W’ has to be in lowercase

Section 3.4

Spacing is missing between the numbers and the units

Pg 8 Line 5: ‘Nano’ has to be ‘nano’ (This applies to all in the text)

Section 3.5

Pg 8 Line 10-12: ‘Because….of films [30]. It is a contradiction since the not all mechanical properties were improved.

Again where the values of tensile strength comes from, since the Stress-strain curves in Figure 5 clearly indicate that tensile strength of PVA is hardly 30 MPa but according to Table 1is 500 MPa.

Section 3.7 page 13.

Line 14-16: How does the results are in accordance with ref [34]? Please elaborate

Conclusion

Pg 12: replace ‘cosmetically’ with ‘commercially’

Reviewer 3 Report

The manuscript deals with the production of packaging material based on PVA and glass flakes. The produced films had their mechanical and WVTR properties investigated and was tested as package for bread. In general, the text must be improved to facilitate the understanding of the given information. In addition, considering the proposed application of these films, it is deeply recommended to characterize the physicochemical and thermal properties of the produced packaging material.